# Novel Taxol-Derivative, STO-1, Induces Selective Anti-Tumor Immunity and Sustained Remission of Glioblastoma Without Triggering Autoimmune Reactions

**DOI:** 10.3390/cells14211703

**Published:** 2025-10-30

**Authors:** Shubhasmita Mohapatra, Adrian Guerrero, Neha Rahman, Khondoker Takia Zaman, Jing Wu, Callistus Onyeagba, Chanyue Hu, Matteo Pellegrini, Jayaram Vankudoth, Seiya Kitamura, Lauren O’Donnell, Youssef Zaim Wadghiri, Probal Banerjee

**Affiliations:** 1Department of Chemistry, The College of Staten Island (CUNY), Staten Island, NY 10314, USA; shubhasmita.mohapatra@csi.cuny.edu; 2Center for Developmental Neuroscience, The College of Staten Island (CUNY), Staten Island, NY 10314, USA; adrian.guerrero@cix.csi.cuny.edu (A.G.); onyeagbajunior@gmail.com (C.O.); 3Bernard and Irene Schwartz Center for Biomedical Imaging, and Center for Advanced Imaging Innovation and Research and Preclinical Imaging, Division for Advanced Research Technologies, Department of Radiology, New York University Grossman School of Medicine, New York, NY 10016, USA; neha.rahman@nyulangone.org (N.R.); youssef.wadghiri@nyulangone.org (Y.Z.W.); 4CUNY Doctoral Program in Biochemistry, New York, NY 10016, USA; kzaman@gradcenter.cuny.edu; 5Hunter College (CUNY), New York, NY 10065, USA; jing.wu@hunter.cuny.edu; 6UCLA Molecular, Cellular & Developmental Biology, Los Angeles, CA 90095, USA; chanyueh25@ucla.edu (C.H.); matteop@mcdb.ucla.edu (M.P.); 7Department of Biochemistry and Chemical Synthesis Core Facility, Albert Einstein College of Medicine, Bronx, NY 10461, USA; jayaram.vankudoth@einsteinmed.edu (J.V.); seiya.kitamura@einsteinmed.edu (S.K.)

**Keywords:** GBM, immunotherapy, autoimmunity, chemotherapeutic, STAT1, Arginase 1, iNOS

## Abstract

**Highlights:**

We have created a novel Taxol derivative that induces anti-tumor immunity in glioblastoma (GBM) tumors without triggering autoimmune reactions, leading to long-term remission in a syngeneic mouse model of GBM.

**Abstract:**

Reprogramming of macrophages into the inflammatory state (also known as M1) is currently considered as an effective way of eliminating cancer cells, but systemic deployment of this strategy is likely to induce dangerous autoimmune reactions. Consequently, converting immunosuppressive M2-type macrophages into M1 systemically is not a safe and effective therapeutic approach against cancer. Through cleavable covalent linking of curcumin to the chemotherapeutic agent Paclitaxel (Taxol), we have created a novel prodrug (STO-1) that, upon intravenous delivery, selectively reprograms tumor-associated microglia and macrophages (TAMs) and eliminates glioblastoma (GBM) without triggering autoimmunity. Demonstrating its therapeutic efficacy, prolonged treatment of six orthotopic GBM-bearing mice with STO-1 resulted in 67% long-term survival, with three surviving mice exhibiting complete tumor clearance and one displaying minimal residual disease, as confirmed by high-resolution ex vivo T2-weighted MRI 85 days after tumor inoculation. In contrast, the vehicle-treated mice displayed extensive intracranial tumors with edema and hemorrhage. Mechanistically, scRNA-seq analysis indicated induction of multiple M1-associated transcripts (*ccrl2*, *cxcl9*, *ccr2*, *ccl5*) consistent with robust TAMs reprogramming. In striking contrast to the M2⟶M1 reprogramming of TAMs, M1-type macrophages were suppressed in the spleens of STO-1-treated cancer-free mice. Therefore, STO-1 induces selective anti-tumor immunity and GBM elimination without triggering systemic autoimmune reactions.

## 1. Introduction

During the past decade, the tumor microenvironment (TME) has come into serious consideration in the context of cancer treatment. This dynamic environment is profoundly shaped by the presence and action of innate immune cells, specifically macrophages, which infiltrate the GBM site [1]. The GBM microenvironment predominantly contains tumor-promoting Arginase 1 (Arg1)^high^, inducible nitric oxide synthase (iNOS)^low^ tumor-associated microglia/macrophages (TAMs). Such TAMs are also known as “alternatively activated” or “M2-like”. These TAMs also include a small number of tumoricidal, Arg1^low^. iNOS^high^ TAMs (M1) and a few non-activated microglia (M0) [1,2]. Although TAMs could exist in multiple states, expressing various marker proteins and cytokines, the characteristic markers mentioned here, Arg1^high^. iNOS^low^ and Arg1^low^. iNOS^high^, for M2- and M1-like states, respectively, are expected to be useful in designing a therapeutic strategy to combat GBM. This approach involves the reprogramming of the M2-like TAMs in the tumor microenvironment (TME) into the M1-like state to eliminate GBM [3].

Through a series of experiments performed in syngeneic mouse models of GBM and cervical cancer, we have demonstrated that phytosomal or antibody-targeted delivery of the dietary compound curcumin (CC) induces the repolarization of the M2-like TAMs into the predominantly M1-like state. This process activates mechanisms that eliminate both GBM cells and GBM stem cells [4,5,6,7]. Generally, the mechanism of this repolarization involves inhibition of the transcription factor STAT3 (Signal Transducer and Activator of Transcription 3) [2], which is recognized for its role in promoting the expression of immune-suppressive cytokines, and also induction of the key enzyme Arg1, which, as mentioned earlier, is a marker for M2-like TAMs [2,7,8]. However, the IL-10 cytokine generally inhibits a second transcription factor, STAT1, by suppressing its expression and phosphorylation [2,7,8]. Consequently, inhibiting STAT3 would lead to STAT1 induction and activation, thereby triggering subsequent STAT1-mediated processes, such as increased expression of iNOS (or NOS2), and IL-12, which are characteristic markers of M1-like microglia and macrophages.

Experiments conducted by Zhang and coworkers achieved the desired M2→M1 reprogramming in both peripheral as well GBM TME macrophages by deploying in vitro-transcribed (IVT) mRNA for Interferon Regulatory Factor 5 (IF5) and inhibitor of nuclear factor kappa-B kinase subunit beta (IKKβ) encapsulated in biodegradable polymeric nanoparticles (NP) in an injectable formulation [9]. The IVT IF5/IKKβ NP formulation was effective in reprogramming cultured human macrophages. Additionally, application of the IVT IF5 NP formulation was effective in improving radiotherapy outcomes in PDGFβ-driven glioma in an RCAS-PDGF-B/Nestin-Tv-a; Ink4α/Arf−/−; Pten−/− transgenic mouse model. Similarly, the possibility of manipulating Colony-stimulating factor 1 receptor (CSF1R) was viewed as a strategy of reprogramming the macrophages in cancer therapy [10,11]. Analogously, De Boeck and coworkers demonstrated that suppression of the glioma-released, TME-altering factor IL-33 stunts recruitment of peripheral innate immune cells and glioma progression [12]. Aligned with earlier findings, inhibition of STAT3 with corosolic acid was shown to cause inhibition of GBM cell proliferation in culture [13].

The study conducted by Zhang and coworkers did acknowledge the importance of delivering the IVT IF5/IKKβ NP formulation selectively to the tumor in order to avoid inducing autoimmune effects [9]. We have crafted our therapeutic approach to avoid such autoimmune reactions completely through a calculated choice of biocompatible molecules and strategic synthetic schemes to generate a novel GBM therapeutic (STO-1) that selectively causes M2→M1 reprogramming of TAMs while inhibiting M1 macrophages in the periphery. We were inspired by the ability of the naturally-derived and clinically-used chemotherapeutic agent (CA) Paclitaxel (Taxol, Pac) to inhibit STAT3 as did CC [14]. However, we were also concerned about the antimetabolite nature of Pac, which could trigger elimination of fast-dividing immune cells, thus precipitating immune suppression. Nonetheless, we expected that the simultaneous presence of CC would reprogram the TAMs without eliminating them [4,5,6,7]. Furthermore, it is widely known that CC functions as an anti-inflammatory agent against microglia and macrophages in the absence of cancer [15,16], which would prevent systemic inflammation, thus averting autoimmunity. Based on such information, we hypothesized that covalently linking Pac to the immune-friendly compound CC through a biocompatible linker would create a drug that would be highly potent against GBM with minimal or no immune-suppressive function. Being covalently linked, CC and Pac would enter cells together, to exert anticancer activity, while maintaining an appropriate balance to spare the immune cells. Our subsequent experiments confirmed that STO-1 eliminated cultured mouse GBM GL261 cells at an IC50 that was in the same range as that for Pac. In sharp contrast, the IC50 for CC was about 15–20 μM. Additionally, we observed that STO-1 reprogrammed M2-type TAMs into the M1-type and further bolstered its anticancer activity by recruiting NK and T-cytotoxic (Tc) cells into the GBM tumor. Most notably, the unique feature of STO-1 is that it does not induce autoimmunity, as evidenced by the suppression of M1-like macrophages in the major macrophage-reservoir, the spleen [17,18,19], in cancer-free mice. As for clinical translatability, we have created STO-1 as an intravenously injectable liposomal formulation in dietary lipids. Once injected, it reaches mouse brain within 15 min at a concentration that is suitable for functional activity.

Since the existing immunotherapies have not shown much efficacy in GBM, our initial effort will be focused on developing the STO-1 technology as a monotherapy. However, one of the therapeutic strategies within the toolkit of “standard of care” for GBM is radiotherapy, and, when combined with radiotherapy, the IVT approach of Zhang and coworkers elicited a significant increase in survival of GBM-harboring mice [9]. Therefore, our preclinical studies will also test a combination of STO-1 treatment with radiotherapy. With this objective, the technology has been licensed to a commercial entity that is currently preparing to submit an IND for FDA approval to conduct a phase I clinical trial, which will begin by testing the safety profile and pharmacokinetics of liposomal STO-1.

## 2. Materials and Methods

**Animals:** The experiments involved 71 adult male C57BL/6 mice, aged 3 to 5 months. These mice were bred within the College of Staten Island (CSI) Animal Care Facility and housed under a controlled 12 h light/dark cycle, with ad libitum access to food and water. All animal-related procedures conducted in this study adhered to the ethical guidelines set forth by the Institutional Animal Care and Use Committee (IACUC) (Approved Protocol# 11-008) at the College of Staten Island.

**Cell culture:** GL261 mouse glioblastoma cells were cultured in Roswell Park Memorial Institute (RPMI-1640) medium supplemented with 10% fetal bovine serum (FBS) and 1% gentamicin. Prior to and during drug exposure, the cells were maintained in serum-free RPMI medium enriched with 1% insulin-transferrin-selenium (ITS) supplement and 1% gentamicin [4].

**Implantation of Cancer Cells in Mice:** GL261 mouse glioblastoma cells (2 × 10^4^ or 10^5^) were implanted stereotaxically in the right forebrain of C57BL/6 mice, following our earlier reports [4,5,7,20]. The GL261-implanted mouse model has been thoroughly characterized through magnetic resonance imaging (MRI) and numerous biological studies [5,7,20,21,22,23,24,25,26].

**Determination of IC50 Using WST-1 Assay:** Cellular proliferation and viability was measured by the WST-1 assay. Briefly 3000 cells/well were plated in each well of a 96-well plate and incubated overnight under suitable culture condition of 5% CO_2_ at 37 °C. An initial 40 mM stock solution of Paclitaxel (Pac) or Paclitaxel–curcumin (Pac-CC, STO-1) in DMSO was diluted to 1 mM by adding to RPMI medium in increments with vigorous mixing. This 1 mM stock was subsequently serially diluted in RPMI medium supplemented with 1% insulin–transferrin–selenium (ITS) (ThermoFisher Scientific, Springfield Township, NJ, USA) to obtain drug concentrations 500 nM and lower (Appendix A). Cells in each well were treated by replacing the medium with RPMI-ITS containing each drug concentration in triplicate, and then incubated for 96 h in a humidified, 5%–CO_2_ incubator. The cells were subjected to intermittent inspection using light microscopy for signs of degeneration, including membrane blebbing and degradation. Subsequently, after 96 h of incubation, the medium in each well was removed, and the cells were gently rinsed three times with PBS. Next, the cells were incubated at 37 °C for 45 min with a 10% (*v*/*v*) solution of WST-1 (Clontech, Mountain View, CA, USA) diluted in serum-free RPMI, as previously detailed in our earlier publication [2]. Optical density was measured at 440 nm using a microplate reader. All experiments were carried out at least three times to determine the IC50 of the drugs for the brain tumor cell lines. GraphPad Prism 10.6 was used for graphical analysis [4,5,6,7,27].

**Preparation of Lipid-complexed STO-1 and Paclitaxel for Intravenous Administration:** STO-1 or Pac was first dissolved in DMSO to create a concentrated stock solution. Subsequently, this stock solution was gradually introduced into sterile PBS in increments with vigorous agitation, resulting in a final concentration of 5% DMSO in PBS. To prepare the liposomal preparations, a soy phospholipid mixture (#541601G, Avanti Research, Alabaster, AL, USA) equal to four times the mass of the drug was added to the drug solution such that a 200 μL aliquot of the final lipid complex would contain 0.46 μmole (0.6 mg for STO-1) of each drug, which would be injected into each mouse. A similarly-prepared but drug-free liposomal preparation in PBS containing <1% DMSO was named as “Vehicle”. The resulting mixture was shielded from light and subjected to three cycles of robust sonication at a medium setting for 10 min bursts with 5 min of intermediate cooling on ice between each sonication cycle [28,29]. This process yielded a consistent emulsion, which was then divided into 200 μL aliquots each containing 0.46 μmole of STO-1 or Pac. Each aliquot was thoroughly purged with nitrogen and stored in a light-protected environment at −20 °C.

**Drug Dosing for Mice Based on Clinically Used Human Dose for Paclitaxel:** Pac alone is mainly used for peripheral cancers in humans, such as breast cancer, and the commonly used dosage is 175 mg/m^2^ (https://reference.medscape.com/drug/taxol-paclitaxel-342187, accessed on 23 October 2023). For a 70 kg human who is 172 cm (5′8″) tall, this value converts to approximately 0.15 mg (0.18 μmol) of Pac per 30 g mouse, but further corrections are needed for the higher metabolic rates in mice [30]. Based on this information, we chose to use 0.46 μmol of Pac or STO-1.

**Treatment of Animals:** Male C57BL/6 mice (3–5 months) received orthotopic implantation of GL261 cells in the right forebrain. **For the long-term (survival) study**, twenty-three mice were implanted with **2 × 10^4^** GL261 cells on day 1, and then, on day 5, six randomly selected mice were used for STO-1 treatment and eleven such mice were used for vehicle treatment. Intravenous treatment was conducted with Vehicle (200 μL of drug-free lipid vesicles) or STO-1 (0.46 μmole in 200 μL of lipid vesicles) or Pac (0.46 μmole in 200 μL of lipid vesicles) every 72 h starting from day 5. This treatment frequency was based on the in vitro doubling time of the GL261 cells, which is about 48 h. Since apoptosis of cells occurs during the cell cycle, we expected that optimal GBM cell death initiated by each STO-1 treatment and TAM-mediated apoptosis should be complete before 72 h. Subsequently, a new STO-1 treatment at 72 h will start the process of apoptosis again. The humane endpoint for this long-term experiment was when a mouse stopped eating and moving and assumed a hunched position. Weight-loss was less than 15% of the original. At this point, the mouse was placed under deep anesthesia and, for MRI scanning, perfused through the heart with PBS followed by 4% paraformaldehyde, followed by decapitation. All perfused mice were decapitated, and the heads were stored in 4% paraformaldehyde. The four (out of six) STO-1-treated mice that had achieved long-term remission were similarly euthanized on day 85 and then decapitated. Each head was stored in a labeled tube until further processing for ex vivo MRI. The heads from three out of the eleven Vehicle mice and all four rescued STO-1-treated mice were subjected to MRI scanning. The brains from the Pac-treated mice that reached humane end points between days 17 and 24 were not subjected to MRI.

Each male mouse **in the short-term-treatment group** was implanted with **10^5^** GL261 cells on day 1. For the 5-day treatments, each drug injection contained 0.46 μmole of the drug in lipid vesicles. The “Vehicle mice” received 200 μL of drug-free lipid vesicles. Three of the GL261-implanted male mice in each of the Vehicle and Pac groups and four for the STO-1 cohort were used. Daily treatment commenced on day 12. Over the course of these 5 days, the mice received intravenous (i.v.) treatments of 200 μL of vesicular STO-1, Pac, or Vehicle per mouse every 24 h. On the fifth day (day 16) of the short-term treatment, the mice also received intranasal delivery of Dylight-CD68 Ab for near-IR scanning, as previously reported [4,20]. Subsequently, on the sixth day (day 17), mice from all three groups were sacrificed, and their brains were near-IR scanned using an Odyssey scanner (Licor) as a second mode of identifying GBM tumors [4,6,20]. Subsequently, the tumors were processed for FC, scRNA-seq, and IHC as described in the later sections.

**Ex vivo MRI:** At the experimental endpoints, mice harboring GBM tumors were deeply anesthetized using a cocktail of Ketamine (100 mg/Kg) and Xylazine (10 mg/Kg). They were then perfused through the heart with PBS followed by 4% paraformaldehyde in PBS. After decapitation with a guillotine, the heads were stored in 4% paraformaldehyde in PBS at 4 °C until similar heads from other cohorts were obtained. Four out of six rescued and healthy mice at day 85 were similarly anesthetized, perfused through the heart, decapitated, and their heads stored in 4% paraformaldehyde at 4 °C until ex vivo magnetic resonance imaging (MRI).

Similar to previous studies [31,32], we employed a high-throughput ex vivo MRI approach, enabling simultaneous scanning of multiple samples using extended overnight imaging time. Each whole head was de-skinned, with ears and facial muscles trimmed off, including the removal of the mandible to fit and secure each of the four heads with glue (Krazy Glue; Elmer’s Products Inc., Columbus, OH, USA) within the four sides of the plunger of an 80 mL syringe. The plunger served as a four-compartment divider, with each head secured in one quadrant.

Polyethylene tubing filled with copper-doped water was used to identify the orientation and location of the samples during the MRI scan. A single tubing was placed next to the first sample, while the fourth sample was identified with double-tubing placed in the neighboring quadrant counterclockwise. The plunger was then inserted into the syringe and immersed in Fomblin (Ausimont, Thorofare, NJ, USA), a proton-free perfluoropolyether fluid. Fomblin matches the magnetic susceptibility of biological tissues without producing background signals, significantly improving shimming. Specifically, the dark background created by Fomblin allows the shimming process to focus entirely on correcting the homogeneity of the sample itself, free from interference by the surrounding background. Additionally, susceptibility matching between the tissue and Fomblin reduces magnetic field distortions that typically arise from air–tissue interfaces, further minimizing artifacts. This dual effect ensures clearer and more accurate MRI images while preventing dehydration.

The sealed syringe, with its tip oriented toward the top, was placed in a vacuum chamber to degas the sample for at least two hours, preventing bubbles or air pockets within the skull that could compromise image quality. MRI scans were performed on a Biospec 70/30 micro-MRI system (Bruker, Billerica, MA, USA) equipped with a zero-helium boil-off 300 mm horizontal bore 7-Tesla (7-T) superconducting magnet (300 MHz) based on ultra-shield refrigerated magnet technology (USR). The magnet was interfaced to an actively shielded gradient coil insert (Bruker BGA-12S-HP; OD = 198-mm, ID = 114-mm, 660-mT/m gradient strength, 130-μs rise time) and powered by a high-performance gradient amplifier (IECO, Helsinki, Finland) operating at 300 A/500 V. This setup was controlled by an Avance-3HD console operated under Paravision 7.0 and TopSpin 3.1.

The MRI setup utilized a Bruker circularly polarized (CP) radiofrequency (rf) birdcage resonator, typically used for scanning a mouse body (inner diameter = 40 mm), which conveniently fits the diameter of an 80 mL syringe. A three-dimensional (3D) *T*_2_-weighted Fast-Spin Echo (FSE) pulse sequence, known for its excellent gray–white matter tissue contrast, was used to help delineate the tumor volume via the expected tissue edema, blood hemorrhage, and necrosis typically associated with brain tumors at an advanced stage. Simultaneous scanning of the four whole heads generated datasets with 150 µm isotropic resolution, where the effective echo time (TEeff) was 60 ms, the echo spacing (ES) was 20 ms, and the repetition time (TR) was 1200 ms. The acceleration factor (AF) was 8, with the number of averages (Nav) being 2. The matrix size (Mx) was 256 × 256 × 256 with antialiasing along the slice phase encoding direction (aa3) of 1.36, resulting in aa3Mx of 256 × 256 × 348. The field-of-view (FOV) was 38.4 mm × 38.4 mm × 38.4 mm, and the bandwidth (BW) was 20 kHz (78.13 Hz/pixel), resulting in an overall imaging time of 7 h and 27 min.

Following completion of the acquisition, all datasets were processed and converted into NIfTI file format to facilitate seamless portability, visualization, and post-analysis using Fiji (version 6). The 150 µm isotopic resolution facilitated 3D image registration and eased analysis of the tumor volume quantification and rendering using the commercial software Amira (version 2024.1, Thermo Fisher). The difference in tissue contrast between the tumor and the surrounding brain tissue enabled precise segmentation and isolation of the tumor mass within the whole brain. This image analysis also enabled 3D segmented rendering of the tumor within the whole head’s anatomical context.


**Flow Cytometry of Immunostained Brain Tumor Cells:**


The brains from the GL261-implanted mice were extricated without fixing (after anesthesia) and rinsed twice with PBS to remove blood cells from the brain. Subsequent removal of each tumor and processing was carried out on the tissue from three animals for each treatment condition. Enzymatic dissociation of each tumor was performed to yield a single-cell suspension, using a Mouse Tumor Dissociation Kit (Miltenyi Biotec, Auburn, CA, USA). The cells were then filtered first through a 70 μm sieve to eliminate clumps and then through a 30 μm sieve to obtain cells with >95% viability. The cells were subsequently fixed using freshly prepared 4% paraformaldehyde (PFA), counted, and preserved in PFA at 4 °C. Approximately 2 million fixed cells from each animal were rinsed with PBS and used for immunostaining. Following each antibody treatment, cells were pelleted and resuspended in washing buffers. Staining was carried out with antibodies against Iba1 (C20) (1:200), iNOS (1:200), and Arg1 (1:200). Secondary antibody-treated sections were employed to establish the threshold [4,5,6,7,23,27,33].

Flow cytometry analysis was conducted using an Accuri C6 flow cytometer (BD). For each sample, gating was applied to select events with forward and side scatter above 10^4^ and 10^3^, respectively, and 100,000 gated events were subjected to analysis. Compensation for fluorescent events was performed to minimize spillover from each channel. Subpopulations of cells were distinguished in different quadrants based on their fluorescence intensities. Events originating from autofluorescence and non-specific fluorescence in the “secondary antibody only” samples were observed as a distinct population in the lower-left quadrant (LL quadrant). Double-stained fluorescent events corresponding to Arg1+, Iba1+ or iNOS+, Iba1+ cells were identified as subpopulations in the upper right (UR) quadrant within the coordinates of 520 nm (green for ARG1 and iNOS) (FL1-A) and 580 nm (red for Iba1) (FL2-A). Mean fluorescence intensity was determined for each double-stained cell using the mean method of statistical analysis [4].


**Single-cell RNA Sequencing:**


Directly after tumor dissociation with gentleMACS™ Tissue Dissociator (Miltenyi Biotec), cell quantity and viability were measured, and a cell suspension was used for Chip loading with 5000 target cells. Preparation of gel beads in emulsion and libraries was performed with Chromium iX Controller and Single-Cell 3′ Gene Expression V3.1 Dual Index Reagent Kits (10X Genomics) at the Epigenetics Core of the Advanced Science Research Center at the City University of New York, according to the User Guide provided by the manufacturer. Library quality and quantity were verified with High-Sensitivity D5000 and High-sensitivity D1000 tapes on a Tapestation 4200 (Agilent Technologies, Santa Clara, CA, USA). Sequencing was then run in the S4 flow cell and paired-end sequenced on a Novaseq (Illumina, San Diego, CA, USA).

**Single-cell RNA-seq Data Processing and Normalization:** Raw sequencing data (BCL files) were demultiplexed and converted to Fastq files using the Cell-Ranger v3.0.1 (10x Genomics) (https://www.10xgenomics.com/support, accessed on 23 October 2023) and bcl2fastq v2.20.0.422 (Illumina). Sequencing results were mapped to a mouse genome GRCm38 (mm10) acquired from the 10x Genomics website and quantified, using a Cell Ranger v.3.0.1. After demultiplexing, data were loaded into Seurat (5.0.3) for analysis. Unsupervised clustering was performed using standard Louvain clustering and visualized using Uniform Manifold Approximation and Projection (UMAP) for cell selection purposes. The cluster analysis was performed on dissociated GBM cells from two vehicle-treated (i, iii) and two STO-1-treated (ii, iv) mice. (i) and (ii) show CD11b+ cells, which were further selected for high Arginine catabolism using the marker *ITGAM*. Non-macrophage cells were manually removed during sub-clustering. Dot plots and violin plots were generated to investigate and visualize the M1 polarization state of microglia and macrophages using a list of key transcriptional markers.

**Antibodies and Dilutions and Immunohistochemistry (IHC):** Iba1 (goat IgG) (C20) (sc28530) (1:50), NKp46 (rabbit IgG) (sc-292796) (1:100), anti-STAT1 (rabbit IgG) (sc-592) (1:100), anti-P-Tyr^701^-STAT1 (mouse IgG) (sc-8394) (1:100), and anti-CD8-α (D-9) (mouse IgG) (sc-7970) (1:100) were used. All antibodies were diluted in 2% goat serum and 2% rabbit serum and 0.1% Triton X-100 in PBS (GRT-PBS). Alexa Fluor 488 rabbit anti-goat, Alexa Fluor 568 goat anti-mouse, and Alexa Fluor goat 488 anti-mouse (Invitrogen) (1:1000 dilutions in GRT-PBS) were used. For IHC, an identical set of brains, as shown in Figure 1, with three mice per treatment, were used to collect the tumor from each brain, fix it in 4% paraformaldehyde, and process it following procedures detailed in our earlier publications [4,5,7].

**Pharmacokinetics of STO-1:** To test whether vesicular STO-1 reaches the brain, we administered a single intravenous injection of 1.2 mg of STO-1 in 200 µL of liposomes prepared as described above. Subsequently, the mice (3 per time point) were sacrificed after 15 min, 30 min, and 1 h, and the brains were collected, minced, and 100 mg of the tissue (100 mg) was diluted with water (300 μL), homogenized, and vortexed with 700 μL acetonitrile (ACN). The suspension was centrifuged, the supernatant was separated into a fresh tube, the pellet was mixed with 1 mL of ACN:H_2_O (70:30), vortexed, and centrifuged again to collect the supernatant in the same tube as the first one. This process was repeated, and all three supernatants were pooled in one tube and evaporated under a stream of nitrogen. The residue was dissolved in 100 μL of acetonitrile–water (70:30) and 30 μL of this solution was injected into a C18 reverse-phase HPLC column for analysis using light absorbance at 430 nm (the absorbance maximum for curcumin). Aliquots of a standard 0.1 mM solution of STO-1 dissolved in acetonitrile–water (70:30) were injected similarly into the column to construct a calibration curve. The mobile phase was a 30–70% gradient of acetonitrile in water containing 0.1% trifluoroacetic acid [34].

## 3. Results

### 3.1. Synthesis of STO-1

In generating a reversibly linked adduct of Paclitaxel (Pac) and curcumin (CC), we followed the approach of creating an esterase-sensitive linkage that is cleaved by intracellular esterases (Figure 2A). We employed the biocompatible glutaric acid linker by first reacting Paclitaxel (**s1**, Figure 2B) with glutaric anhydride and triethylamine in dichloromethane (DCM) under anhydrous nitrogen (see Section 2). Following aqueous workup and extraction of the product (**s2**) into DCM, it was purified by silica-gel column chromatography and then subjected to esterification with CC (**s3**) in the presence of 1-ethyl-3-(3-dimethylaminopropyl carbodiimide (EDC) and dimethylaminopyridine (DMAP) in dichloromethane under anhydrous CC. The resulting product obtained was purified by silica-gel column chromatography and analyzed by NMR, liquid chromatography (LC), and mass spectrometry (MS) to confirm 98% purity (Figure 1A,B).


**Synthesis of Paclitaxel–Curcumin (Pac-CC) (STO-1):**


**5-(((1S,2R)-1-benzamido-3-(((2aR,4S,4aS,6R,9S,12S,12bS)-6,12b-diacetoxy-12-(benzoyloxy)-4,11-dihydroxy-4a,8,13,13-tetramethyl-5-oxo-2a,3,4,4a,5,6,9,10,11,12,12a,12b-dodecahydro-1H-7,11-methanocyclodeca [3,4]benzo [1,2-b]oxet-9-yl)oxy)-3-oxo-1-phenylpropan-2-yl)oxy)-5-oxopentanoic acid (s2) (Figure 2B):** Reaction conditions were adapted from a previously published article [35]. To a solution of protected Paclitaxel (1 g, 4 mmol) and glutaric anhydride (4 g, 0.04 mol) in 100 mL of CH_2_Cl_2_ was added TEA (2 mL, 0.02 mol). Stirred the reaction mixture at room temperature overnight. This mixture was then washed with NaHCO_3_ (10%, 2 × 30 mL), water (2 × 30 mL), and brine (30 mL). The organic phase was then dried (Na_2_SO_4_), filtered, and concentrated under reduced pressure. The crude product was then purified by combi-flash chromatography (DCM/Methanol). Yield: 1.6 g (60%), white solid.


**(1S,2R)-1-benzamido-3-(((2aR,4S,4aS,6R,9S,12S,12bS)-6,12b-diacetoxy-12-(benzoyloxy)-4,11-dihydroxy-4a,8,13,13-tetramethyl-5-oxo-2a,3,4,4a,5,6,9,10,11,12,12a,12b-dodecahydro-1H-7,11-methanocyclodeca [3,4]benzo [1,2-b]oxet-9-yl)oxy)-3-oxo-1-phenylpropan-2-yl (4-((1E,6E)-7-(4-hydroxy-3-methoxyphenyl)-3,5-dioxohepta-1,6-dien-1-yl)-2-methoxyphenyl) glutarate (s4) (Figure 2B):**


To a solution of compound **s2** (350 mg, 362 μmol) and compound **s3** (133 mg, 362 μmol) in 30 mL methylene chloride was added EDC (398 mg, 2.08 mmol) and DMAP (33.8 mg, 277 μmol) at ambient temperature, the reaction was allowed to stir for 12 h. The reaction mixture was concentrated under vacuum and purified via preparative HPLC using a H_2_O-CH_3_CN gradient (99:1 to5:95, *v*/*v*, 0.1%FA). A fraction containing the target molecule was lyophilized to give a yellow solid. (100 mg, 21%). ^1^H NMR (600 MHz, CDCl_3_) δ 8.16–8.14 (m, 2H), 7.74–7.72 (m, 2H), 7.64–7.59 (m, 3H), 7.52 (d, J = 7.9 Hz, 2H), 7.48–7.45 (m, 1H), 7.43–7.37 (m, 4H), 7.33 (d, J = 8.0 Hz, 3H), 7.14 (ddd, J = 12.4, 8.2, 1.9 Hz, 2H), 7.07 (dd, J = 9.3, 1.9 Hz, 3H), 7.02 (d, J = 8.2 Hz, 1H), 6.94 (d, J = 8.2 Hz, 1H), 6.55 (d, J = 15.8 Hz, 1H), 6.50 (d, J = 15.8 Hz, 1H), 6.31 (s, 1H), 6.30–6.26 (m, 1H), 6.02 (dd, J = 9.2, 3.0 Hz, 1H), 5.89 (s, 1H), 5.83 (s, 1H), 5.69 (d, J = 7.1 Hz, 1H), 5.54 (d, J = 3.0 Hz, 1H), 5.00–4.98 (m, 1H), 4.46 (td, J = 6.6, 3.3 Hz, 1H), 4.32 (d, J = 8.5 Hz, 1H), 4.22–4.20 (m, 1H), 3.95 (s, 3H), 3.83 (d, J = 7.0 Hz, 1H), 3.77 (s, 3H), 2.74–2.55 (m, 7H), 2.49 (d, J = 4.0 Hz, 1H), 2.46 (s, 3H), 2.41–2.37 (m, 1H), 2.23 (s, 3H), 2.21–2.17 (m, 1H), 2.08 (d, J = 7.1 Hz, 2H), 1.97 (d, J = 1.5 Hz, 3H), 1.92–1.87 (m, 1H), 1.69 (s, 3H), 1.25 (s, 3H), 1.14 (s, 3H). ^13^C NMR (151 MHz, CDCl_3_) δ 203.8, 184.5, 181.7, 171.9, 171.2, 170.8, 169.8, 168.0, 167.1, 167.0, 151.2, 148.0, 146.8, 142.8, 141.2, 141.0, 139.2, 136.9, 134.2, 133.7, 133.5, 132.8, 131.9, 130.2, 129.1, 129.0, 128.7, 128.6, 128.4, 127.5, 127.1, 126.4, 124.4, 123.1, 123.0, 121.7, 120.9, 114.8, 111.4, 109.6, 101.5, 84.4, 81.1, 79.2, 75.6, 75.1, 74.1, 72.1, 71.8, 58.5, 55.9, 55.8, 52.6, 45.5, 43.2, 35.5, 35.5, 32.6, 32.5, 29.7, 26.8, 22.7, 22.1, 20.8, 20.2, 14.8, 9.6. MS: *m*/*z* calculated for C_73_H_76_NO_22_ [M + H]+: 1318.49, found: 1318.40, Purity (HPLC-UV 254 nm): >95%. (^t^R = 3.791 min).

### 3.2. STO-1 Crosses the Blood–Brain Barrier (BBB)

Following tail-vein infusion of STO-1 in liposomes (see Supplemental Information) [36], brains from mice were harvested after specific time intervals, homogenized in 70% aqueous acetonitrile (ACN), and the extracts analyzed by high-performance liquid chromatography (HPLC) calibrated against a standard STO-1 curve. Within 15 min from infusion, STO-1 was detected in the brain at approximately 400 nM and was metabolized over time (Appendix A).

### 3.3. STO-1 and Pac Have Similar Potency in Eliminating Mouse Glioblastoma GL261 Cells In Vitro

We evaluated the efficacy of STO-1 in eliminating GL261 cells through IC50 analysis. STO-1 exhibited an IC50 of 20.51 nM, which was not statistically different (*p* = 0.42) from that for Paclitaxel alone (46.82 nM) (Appendix A).

### 3.4. Short-Term Treatment of GBM Mice to Study the Tumor Microenvironment (TME) in Vehicle-, Pac- and STO-1-Treated Groups

As a proof of our concept that STO-1 elicits an M2→M1 reprogramming of tumor-associated microglia and macrophages (TAMs), we generated GBM tumors and then subjected the GBM-harboring mice to a short-term daily vehicle or drug treatment. To ensure tumor persistence at the time of analysis, we implanted 10^5^ GL261 cells on day 1 and monitored the effect of daily *i.v.* administration of liposome-complexed [36] Vehicle, STO-1 (0.46 μmole), and Pac (0.46 μmole) from day 12 for five days on the GBM tumor, TAMs, and the TME. On day 17 post-implantation, mice were deeply anesthetized and euthanized (Figure 3A). Liposomal STO-1 or Pac was each prepared by sonicating with four times the mass of a soy phospholipid mixture in such a way that 0.46 μmole of each drug was contained in 200 μL of liposomes in phosphate-buffered saline (PBS) for each injection. The Vehicle contained drug-free liposomes (200 μL) (see Section 2).

Tumors from several cohorts of mice were excised and processed for downstream analysis: dissociation into cells for flow cytometry (FC) and single-cell-RNA sequencing (scRNAseq), or fixation for IHC (Figure 3B). A separate cohort of brains was fixed and subjected to ex vivo MRI sand 3D reconstruction to visualize GBM tumors (Figure 3C). Figure 3D shows a representative unfixed brain with hemorrhagic GBM tumor, which was resected and processed as outlined in Figure 3A,B.

Each tumor was identified by its hemorrhagic appearance (Figure 3D), enabling precise dissection from adjacent non-tumor tissue. The isolated tumor tissue was then either fixed for histological analysis or dissociated for scRNA-seq and flow cytometry (FC).

**Sharp Induction of iNOS (M1 marker) and Inhibition of Arg1 (M2 marker) in Iba1+ TAMs in the STO-1-treated Mice.** In flow cytometric analysis, compared to Vehicle (empty liposomes)-treated mice, we observed a marked decrease in Arg1+ M2-like TAMs (Iba1+) and a sharp increase in iNOS+ (M1-like) TAMs in liposomal STO-1-treated mice (Figure 4A–C). Less pronounced changes were observed in TAMs from liposomal Pac-treated mice.

Tumors from an identical cohort as shown in Figure 3, with three mice per treatment group, were subjected to IHC analysis. The data strongly supported the FC results: tumors from the liposomal STO-1-treated mice showed a significant decrease in Arg1 and a significant increase in iNOS expression in the Iba1+ TAMs compared to Vehicle-treated mice (Figure 5A–C). Although less prominent, a similar trend was also observed in Pac-treated mice.

### 3.5. Dramatic Inhibition of STAT3 and P^705^-STAT3 and Induction/Activation of STAT1 in Iba1+ TAMs in STO-1-Treated GBM Tumors

Further IHC analysis was conducted on tumors from Figure 3 to assess the expression of STAT3 (M2-linked) and STAT1 (M1-linked) in liposomal Pac and STO-1-treated GBM. We observed a dramatic inhibition of both active STAT3 (P-Tyr^705^ STAT3) and total unphosphorylated STAT3 in mice treated with either Pac or STO-1 (Figure 6A–C). Notably, STO-1 treatment also led to increased activation (phosphorylation of STAT1 at Tyr^701^) and elevated expression of STAT1 compared to Vehicle-treated mice. In contrast, Pac treatment caused less prominent STAT1 induction and no significant activation (Figure 6D–F). Based on data presented in Figure 5 and Figure 6, we proposed a preliminary mechanism (Figure 6G). However, a specific observation suggests additional complexity in STO-1’s anti-GBM activity.

While Pac-mediated inhibition of both phosphorylated and unphosphorylated STAT3 was comparable to that caused by STO-1, Pac’s effect on STAT1 induction and activation was weaker. This suggests that STO-1-induced STAT1 activation may not solely result from STAT3 inhibition, potentially explaining its stronger M2→M1 repolarization effect (Figure 6).

### 3.6. Single-Cell RNA Sequencing

For single-cell RNA sequencing (scRNA-seq) analysis, we selected dissociated cells expressing CD11b (*ITGAM*), a marker for microglia, macrophages, and monocytes (Figure 7A,B). During sub-clustering, we examined gene markers in clusters 2, 4, 6, 9, 14, and 20, identifying high *ITGAM* expression and excluding clusters 9 and 20 as likely neutrophils.

We generated dot plots and violin plots to visualize expression levels of *Arg1*, *NOS2*, and *STAT1*. Interestingly, mRNA levels from scRNA-seq of *Arg1* and *iNOS* (*NOS2*) did not correlate with protein levels of Arg1, iNOS, and STAT1 observed in FC and IHC (Figure 4, Figure 5 and Figure 6D). This aligns with the literature showing that mRNA–protein concordance is only 46 and 68% [42,43,44], due to post-transcriptional and translational regulations. Thus, the observed Arg1^high^, iNOS^low^ M2-type to Arg1^low^, iNOS^high^ M1-type TAM switch is biologically “real”, even if it is not reflected at the mRNA (transcription) level.

Nonetheless, our scRNA-seq revealed that TAMs from STO-1-treated mice expressed higher levels of M1-associated transcripts, including *ccrl2*, *cxcl9*, *ccr2*, *ccl5*, and *rel* compared to Vehicle-treated mice. Notably, ccrl2 expression was higher in STO-1-treated TAMs than in Pac-treated mice (Figure 7G).

### 3.7. Recruitment of Activated NK and CD8+ Tc Cells in the Tumors of STO-1-Treated Mice

Consistent with our earlier findings in CC-treated GBM mice [7], we observed a significant influx of activated (NKp46+, tumoricidal) NK cells in STO-1-treated tumors (Figure 8A,B), with fewer NK cells in Pac-treated tumors. As previously reported, NK cell recruitment enhances M2→M1 TAM polarization [7]. IHC analysis of tumor sections from Figure 3 also showed a robust recruitment of CD8+ cytotoxic Tc cells into the TME in STO-1-treated mice (Figure 8C,D), with much lower levels in Pac-treated mice.

### 3.8. STO-1 Treatment Does Not Trigger Autoimmune Effects in Normal Tissue

While the induction of M1-like TAMs, NKp46+ (activated) NK cells, and CD8+ (cytotoxic) Tc cells in the TME is beneficial for GBM tumor clearance, such inflammation in normal tissues could be harmful for a living organism. To test this possibility, we examined whether systemic (i.p.) STO-1 treatment induced M1-like macrophages in the spleens of cancer-free mice. Unlike lipopolysaccharide (LPS) treatment, which caused a strong M1-like macrophage induction, STO-1 treatment caused no such induction—in fact, it led to reduced iNOS expression in Iba1+ splenic macrophages (Figure 8E,F).

Therefore, STO-1 is a highly effective anti-GBM agent that induces a potent inflammatory response within the GBM tumor while sparing normal tissues, suggesting minimal risk of systemic autoimmunity.

### 3.9. Prolonged STO-1 Treatment Induces Long-Term Remission of GBM

In our studies, we used the well-characterized, immunocompetent mouse model of GBM, established through orthotopic implantation of GL261 mouse glioblastoma cells in C57BL/6 mice [22,23,26,45]. Using this model, we demonstrated that STO-1 treatment induces long-term remission in GBM-implanted mice (* *p* = 0.01) (Figure 9A,B). Ex vivo magnetic resonance imaging (MRI) of brains from Vehicle-treated mice revealed the presence of large GBM tumors (Figure 9C–E). In contrast, MRI scans of STO-1-treated mice 85 days post-implantation showed that three out of four surviving mice were tumor-free, while one mouse exhibited a small residual tumor (Figure 9F and Figure 10).

### 3.10. Prolonged Pac Treatment Does Not Rescue Mice from GBM

In sharp contrast to STO-1-induced rescue of mice from GBM, prolonged liposomal Pac treatment did not increase survival time for the GBM mice compared to the Vehicle-treated mice (Appendix A). Rather, Pac treatment had a significantly negative effect on the survival of GBM mice.

## 4. Discussion

General cancer therapy during the past several decades has involved a single-target approach aimed at eliminating rapidly dividing cancer cells [46]. The past two decades have ushered in a new strategy, hailed as “immunotherapy”, involving reversal of a dampening effect that cancer cells unleash through cell-surface PD-L1 molecules, which bind to PD-1 molecules expressed by the T-cells to silence these adaptive immune cells [46]. Similarly, ex vivo activation of patient-derived T-cells using the CAR T-cell technology followed by infusion of the engineered T-cells has also yielded some success in a few types of cancer but not GBM [47,48,49].

The GL261-implanted syngeneic mouse model, which has been extensively characterized in prior studies [22,23,26,45], has proven invaluable for studying glioblastoma and its interaction with the immune system. It reliably produces an immunosuppressive microenvironment that closely mimics key aspects of human GBM, making it a robust platform for evaluating novel immunotherapies and understanding immune-mediated tumor control. The GL261 cell line harbors a K-Ras mutation, which is not observed in human GBM. However, the *K-Ras* mutation, along with mutations of the tumor suppressor *p53* gene result in overexpression of the *c-myc* gene [22]. Three research teams have reported similar aberrations in human gliomas [50,51,52]. As partially immunogenic, GL261 tumors express MHC I at high levels, but they express no MHC II, and the co-stimulatory molecules B7-1 and B7-2 required for T-cell activation [22]. It is expected that the lack of these molecules in human GBM cells enables them to evade immune surveillance [53,54,55].

The GL261 cells are IDH-wildtype, whereas an IDH mutation is observed in a subset of human GBM patients. To investigate the immunogenic effect of this mutation, Pellegatta and coworkers [56] modified the GL261 cells to express the IDH R132H mutation and generated mouse GBM in mice. Subsequently, by using five discrete peptides covering the IDH1 mutation region to immunize these mice, they could prolong the survival of the mice. This highlighted the GL261 model’s adaptability for investigating the immunogenicity of specific genetic alterations [56].

In reference to other studies on the innate immune system, the stimulation of innate immune cells often involves the cyclic GMP–AMP synthase (cGAS)–stimulator of interferon genes (STING) pathway. To study the involvement of this pathway in the GL261 GBM model, a recent publication is of importance [57]. Thus, Berger and coworkers employed a STING agonist, ADU-S100, to stimulate the innate immune cells in the TME of GL261 mouse tumors, which increased long-term survival. They also showed that depletion of the innate immune NK cells eliminated the ADU-S100-evoked prolonged survival. Thus, the findings of this study are consistent with our observations demonstrating the involvement of NK cells in GBM elimination.

Collectively, such studies strongly suggest that the use of innate immune cells to combat cancer could be effective, and, consequently, other strategies to reprogram innate immune cells, such as TAMs, from the M2 to the M1 state are currently being explored [58,59,60,61]. In one of these studies, Baumgartner and coworkers used a small molecule inhibitor to selectively and potently inhibit protein tyrosine phosphatases PTPN2 and PTPN1, which are believed to be key regulators of inflammation. This caused anti-tumor immunity against cancers that are refractory to PD-1 blockade [61]. Similarly, Pyonteck and coworkers used a BBB-penetrant colony stimulating factor-1 receptor (CSF-1R) inhibitor BLZ945 with an IC50 of 1 nM to reprogram macrophages into the inflammatory M1 state and block mouse glioma progression [62]. Such approaches are valuable, but they also raise some concerns. For example, the nearly total inhibition of PTPN2, PTPN1, and CSF-1R may adversely affect healthy cells that require these functional proteins for normal survival. This concern applies to both T-cell-based approaches and the pharmacological approaches of inhibiting some signaling proteins that have systemic importance. Furthermore, creating an inflammatory state in an entire organism is likely to precipitate autoimmune episodes that could be life-threatening [63].

Most notably, STO-1 creates an inflammatory TME through reprogramming of TAMs but without causing a systemic M2⟶M1 switch in macrophages. To further strengthen the anti-tumor activity, STO-1 treatment evokes intra-tumor recruitment of activated and tumoricidal NK cells and CD8+ cytotoxic T-cells (Figure 6) [4,5,6,46,64]. Thus, STO-1 stands as a prototypic molecule that offers the proof of our concept of turning only the TME against the GBM tumor. Furthermore, because TAMs and NK cells do not require antigen presentation, our strategy will be effective in eliminating GBM cells and GBM stem cells regardless of their mutation load.

STO-1 alone is as potent as Pac in directly killing cultured GL261 cells (Appendix A). As an anti-GBM agent, STO-1 caused long-term remission in syngeneic mice harboring GBM tumors (Figure 9A–F). In striking contrast, similar long-term treatment with liposomal Pac did not cause extended survival of GBM mice beyond that observed for the Vehicle-treated mice (Appendix A). This observation suggests that Pac functions mainly as a proliferation blocker, which is expected to arrest the proliferation and anti-tumor function of the innate immune cells. As a protector of the immune cells [4,7], the presence of CC brings about a dramatic functional change, thereby creating a powerful anti-GBM agent (STO-1) that selectively creates an inflammatory state within the tumor but without triggering systemic inflammation.

Our experiments also reveal an important, but somewhat unappreciated aspect of gene expression. In our published and current studies, we have demonstrated strategies for reprogramming the GBM-associated Iba1+ TAMs from the Arg1^high^, iNOS^low^ tumor-promoting M2-like state to the tumoricidal Arg1^low^, iNOS^high^ M1-like state. Our rigorous studies have demonstrated this phenomenon repeatedly at the protein level. However, a rigorous scRNA-seq analysis of *ITGAM*+ cells dissociated from the tumors shown in Figure 3 failed to reveal a parallel suppression of *Arg1* and induction of *iNOS* at the message level (Figure 7). Nonetheless, our scRNA-seq did reveal induction of several other messages linked to the anti-tumor M1 state of TAMs in the STO-1-treated mice, such as *ccrl2*, *cxcl9*, *ccr2*, *ccl5*, and *rel*. Additionally, the TAMs in the STO-1-treated mice showed a higher induction level of the *ccrl2* gene compared to those in the Pac-treated mice (Figure 7). Although it is clear that the expression of any protein is regulated at both the transcriptional and translational levels, many published studies often predict complete signaling axes based solely on scRNA-seq, which only measures message (mRNA) levels. Yet, the concordance between mRNA and protein varies between only 46 and 68% [42,43,44]. A likely reason is that regulation of protein expression occurs in multiple steps in cap-dependent eukaryotic translation [65,66,67]. We observed that the changes in the expression of Arg1, iNOS (NOS2), and STAT1 are not displayed in scRNA-seq at the mRNA level. As mentioned earlier, concordance between mRNA and protein levels is much less than 100%. This invokes the possibility that the regulation of these crucial proteins may be occurring at the translational level. In fact, the phenomenon that both Pac and STO-1 inhibit STAT3, but only STO-1 induces STAT1 suggests that this regulation possibly involves a post-transcriptional mechanism. Thus, a regulation process at the translational level may explain this non-congruence between mRNA and protein levels for Arg1, iNOS (NOS2), and STAT1 in the TAMs. Both deregulation as well as dysregulation of crucial proteins at the translational level has been reported in many types of cancer and a number of diseases (reviewed in [68]). Multiple research teams have shown the involvement of modified mRNA, tRNA, translation factors, ribosomes or regulatory elements (functioning at the 5′- and also 3′-untranslated regions of the mRNA) in such regulation [69,70,71]. It is noteworthy that GBM, which is classified as a “cold tumor” based on its immunosuppressive interior, uses secreted cytokines and chemokines to recruit microglia and macrophages and convert them into the immunosuppressive and tumor-promoting M2 state. It is possible that this abnormal environment (TME) created by the mutated GBM cells promotes translational mechanisms to exert both dys- as well as deregulation of some proteins involved in the reprogramming of TAMs (Figure 6G). It is expected that the surge in NO and the subsequent reactive nitrogen species formed contribute to the elimination of GBM cells and GBM stem cells [7]. Our future studies will address this possibility at the molecular level in co-culture of both mouse and human GBM cell lines along with mouse and human macrophage cell lines.

Such in vitro studies will also address another observation that sets apart the signaling of Pac and STO-1. As mentioned in the Results section, both Pac and STO-1 were observed to significantly inhibit STAT3 and P-*Tyr^705^*-STAT3 in the TAMs, but only STO-1 appeared to cause a dramatic induction of STAT1 and P-*Tyr^701^*-STAT1 (Figure 6). This strongly suggests that the induction and activation of STAT1 may not be completely linked to the inhibition of STAT3. Currently, no macrophage-selective knockout studies are available for STAT3 or STAT1. Knockout of these proteins in the whole animal will not help us understand this anomalous mechanistic detail. However, in our future studies we will use GBM cell–macrophage co-culture systems and commercially available STAT inhibitors to tease out such mechanisms. Such co-culture studies will also explore the role of GBM cell-released cytokines in the TME in the STO-1-evoked reprogramming of TAMs into the inflammatory M1-type in sharp contrast to its anti-inflammatory effect on splenic macrophages.

From the therapeutic angle, we need to address the question of STO-1 dose versus GBM remission rate in mice. Experiments included here demonstrate the lack of inflammatory responses in spleen macrophages of cancer-free mice. The spleen is the largest secondary lymphoid organ and is pivotal in generating immune responses to blood-borne antigens. It is the reservoir of four distinct populations of macrophages even under normal physiological conditions [19]. It also happens to be the largest blood filter. Splenic macrophages last for months to a year. In contrast, although the number of macrophages in the liver (a.k.a Kupffer cells) are higher in number than the splenic macrophages, each Kupfer cell lasts only for 3.8 days and the primary role of these cells is to clear debris (through phagocytosis) from the circulating portal system that traverses the liver [72,73]. Based on such reasons, we chose to use the splenic macrophages to test the systemic effect of STO-1. Nonetheless, this study will be expanded later to include other immune-responsive organs such as liver, pancreas, and kidneys. Hematoxylin-and-Eosin (H&E) staining will also be conducted to analyze any lesion at the morphological level [4]. Based on the safety profile, future studies will address the possibility of improving GBM remission rates in mice through an increase in treatment frequency or STO-1 dose escalation or STO-1 treatment parallel to radiotherapy [9].

The justification of the use of CC in structurally modifying Paclitaxel invokes questions about anticancer efficacy of CC at the therapeutic level in humans. Successful clinical trials have been conducted with curcumin formulations by several teams of scientists and clinicians. Thus, Basak and coworkers used an FDA-approved, turmeric-derived, curcumin-based formulation, APG-157DS, in a placebo-controlled human trial for oral cancer to demonstrate a reduction in pro-tumor cytokines IL-1β, IL-6, and IL-8 in the salivary supernatant fluid of patients with cancer. Additionally, tumor tissues of subjects demonstrated boosted expression of genes linked to differentiation and T-cell recruitment to the TME [74]. In a second trial of APG-157DS in human patients with head and neck cancer, Tosevska and coworkers used a methodology involving circulating plasma cell-free RNA (cfRNA) as a sensitive readout of patient response to show that APG-157 treatment elicits leukocyte activation and cytokine upregulation in cancer patients but not healthy or placebo-treated patients [75]. In the most recent trial of APG-157DS-treated subjects, 77% achieved pathological responses (23% near-complete, 23% major, 31% partial), whereas 15% had stable disease and 8% displayed progression. Consistent with our observations, APG-157DS elicited a reduction in Ki-67, increased infiltration of CD8+ cells, and reprogrammed macrophages to the M1 phenotype [76]. In yet another clinical trial, Saghatelyan and coworkers used CC in combination with Paclitaxel in breast cancer patients to measure “Response Evaluation Criteria in Solid Tumors” (RECIST), progression-free survival (PFS), time to tumor progression (TTP), time to treatment failure (TTTF), safety, and quality of life [77]. Inclusion of CC caused a significant increase in objective response rate (ORR) compared to placebo (51% vs. 33%, *p* < 0.01) at 4 weeks of follow-up. The inter-group difference increased when only patients completing treatment were considered (61% vs. 38%, *p* < 0.01, odds ratio = 2.64).

The practical aspects of developing a safe anti-GBM agent such as STO-1 are also quite promising. Paclitaxel (Taxol) is one of the least expensive anti-cancer agents. It is widely used for peripheral cancers everywhere, including all developing countries. Curcumin (CC) is a safe dietary agent that is a component of the inexpensive culinary spice turmeric. Several safety studies such as Phase I clinical trials have demonstrated that humans can tolerate up to 8–12 g of CC per day with little or no side effects [78,79,80,81]. Finally, the linker, glutaric acid (Figure 1A), is produced from lysine in the mammalian body and then metabolized through the formation of glutaryl-CoA, which is converted by glutaryl-CoA dehydrogenase to crotonyl-CoA that eventually produces acetyl-CoA molecules for the citric acid cycle [82,83]. Therefore, the novel compound, STO-1, holds immense promise as a safe prodrug that could usher in a new era in affordable GBM therapy.

Although it is generally accepted that most solid tumors recruit innate immune cells—such as macrophages—and program them to the immunosuppressive, pro-tumor M2 state, the cytokine repertoire used by one cancer cell type to accomplish this task may differ from another cancer cell type. Therefore, in addition to the future studies mentioned earlier, it will be pivotal to test the anti-GBM efficacy of STO-1 and its ability to induce the M2⟶M1 reprogramming of the TAMs using a different mouse GBM cell line. Thus, our future studies will use the syngeneic CT2a cells (also developed in the C57BL/6 mice [84]) to generate the GBM tumor.

The STO-1 technology has been licensed to Vascarta, Inc. (Summit, NJ, U.S.A.). This company is currently preparing to include liposomal STO-1 in their repertoire of clinical trials. Their immediate goal is to prepare an IND application for FDA approval before the end of this calendar year. This will be followed by a phase I clinical trial for safety and pharmacokinetics.

## Figures and Tables

**Figure 1 cells-14-01703-f001:**
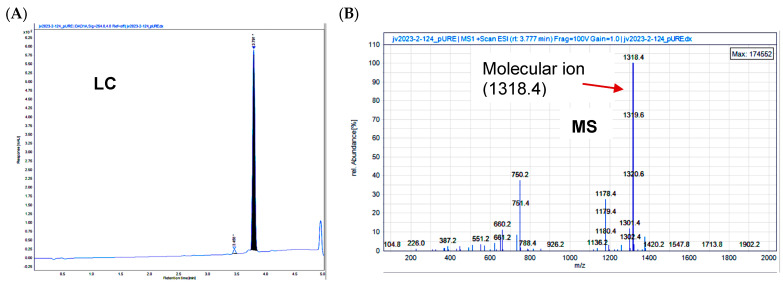
Confirmation of the structure of STO-1 (Pac-CC). (**A**) Liquid chromatography (LC) and (**B**) mass spectrometry (MS) of Pac-CC (STO-1).

**Figure 2 cells-14-01703-f002:**
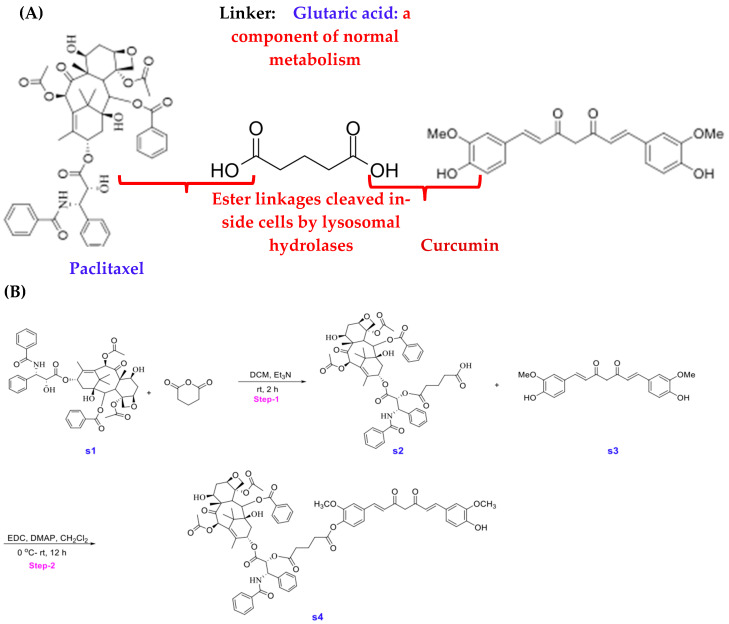
(**A**) Strategy for the creation of Stomalignum-1 (STO-1). (**B**) Experimental steps for the synthesis of Stomalignum-1 (STO-1).

**Figure 3 cells-14-01703-f003:**
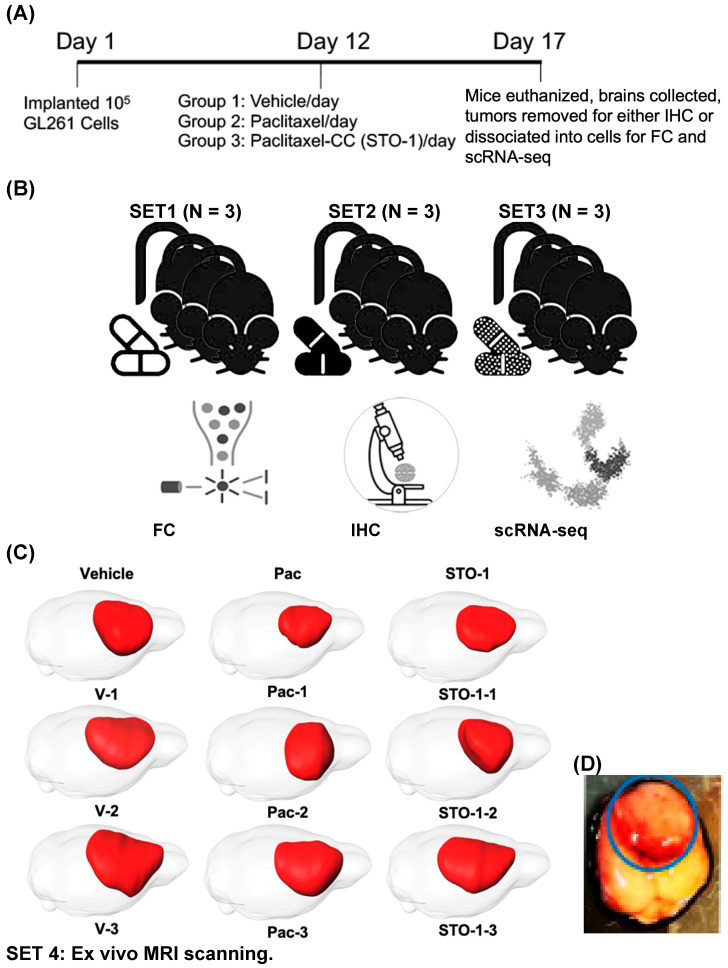
Five-day treatment with Vehicle, Pac, and STO-1 starting on day 12 post-GL261 implantation facilitates M2/M1 analysis of tumor-associated microglia and macrophages (TAMs). (**A**) For each set of nine 4–5-month-old mice, after orthotopic implantation of 10^5^ GL261 cells per mouse on day 1, the mice were randomly divided into three groups. Beginning on day 12, they were subjected to a five-day daily treatment with Vehicle (empty lipid vesicles; *N* = 3), liposomal Pac (0.46 μmole; *N* = 3), or liposomal STO-1 (0.46 μmole; *N* = 3). (**B**) Four identical sets of treated mice were thus created. GBM tumors from two cohorts were dissociated into single cells for flow cytometry (FC) and single-cell RNA sequencing (scRNA-seq), while tumors from the third cohort were fixed for immunohistochemistry (IHC). Thus, each treatment group was represented in analyses by FC, IHC, and scRNAseq, respectively. (**C**) MRI and 3D rendering were performed on brains from the fourth set of nine mice. (**D**) A representative unfixed brain harboring a GBM tumor from a vehicle-treated mouse is shown, from which the highly visible, hemorrhagic tumor (shown in the blue circle) is excised and processed as illustrated in Figure 3A.

**Figure 4 cells-14-01703-f004:**
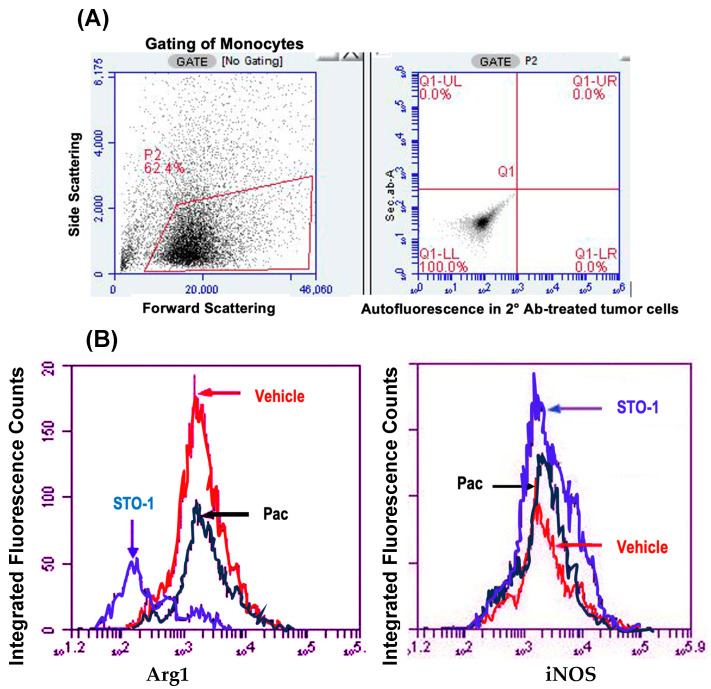
Flow cytometry (FC) of cells dissociated from mouse GBM tumors shown in Figure 3: STO-1 (Pac-CC) treatment inhibits M2-like TAMs and induces M1-like TAMs relative to Vehicle-treated mice. (**A**) Gating of monocytes from dispersed tumor cells, with thresholds for green (horizontal) and red (vertical) fluorescence set based on autofluorescence controls. (**B**) Total fluorescence values from Iba1, Arg-1+,+ cells (left) and Iba1+, iNOS+,+ cells (right) within the gated lymphocyte population (“Integrated fluorescence”) were compared across Vehicle (empty liposomes), liposomal Pac, and liposomal STO-1 treatment groups. (**C**) Quantification of integrated fluorescence values from three mice per treatment (*N* = 3) demonstrates that STO-1-treatment results in a significant decrease in Arg-1 expression and a significant increase in iNOS expression among Iba1+ lymphocytes. Integrated fluorescence = average fluorescence per cell × total number of iNOS, Iba1+,+ or Arg-1, Iba1+,+ lymphocytes (*N* = 3 per treatment; dispersed GBM tumor cells from three brains per group were analyzed in three independent FC runs).

**Figure 5 cells-14-01703-f005:**
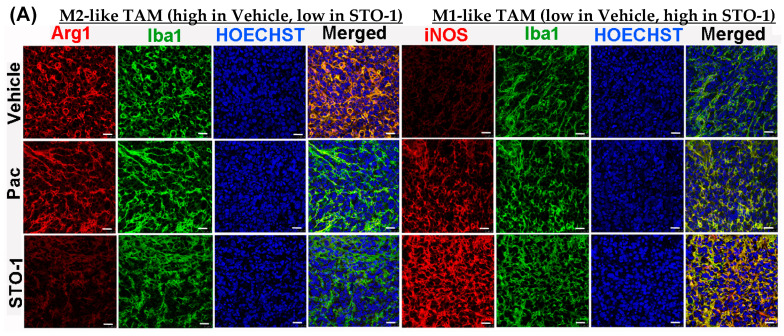
STO-1-mediated repolarization of Iba1+ TAMs from Arg1^high^, iNOS^low^ M2 state to the Arg1^low^, iNOS^high^ M1 state in tumor tissue shown in Figure 3. (**A**–**C**) Tumors from STO-1-treated mice show a dramatic inhibition of Arg1 and a sharp increase in iNOS in the Iba1+ TAMs. (*N* = 3 mice per treatment). Scale bar: 25 μm.

**Figure 6 cells-14-01703-f006:**
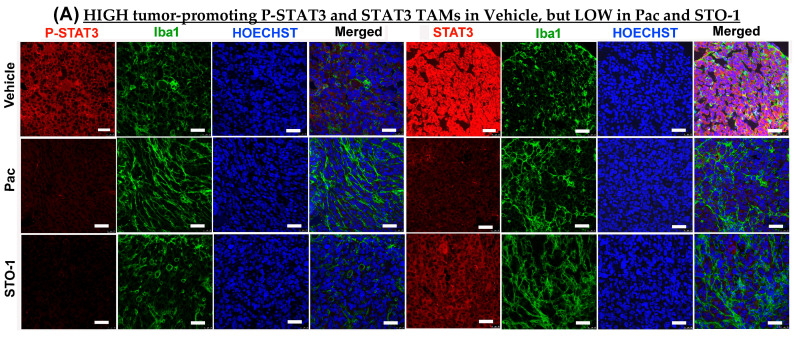
STO-1-mediated inhibition of STAT3 and induction of STAT1 in tumor tissue obtained in Figure 3. (**A**–**C**) P-Tyr^705^-STAT3 (active STAT3) and STAT3 (N = 3 mice per treatment. (**D**–**F**) P-Tyr^701^-STAT1 (active STAT1) and STAT1 (N = 3 mice per treatment). Scale bar: 25 μm. (**G**) A preliminary mechanistic framework: STAT3 inhibition, induction of STAT1, and M2⟶M1 shift in TAMs [7,14,37,38,39,40,41].

**Figure 7 cells-14-01703-f007:**
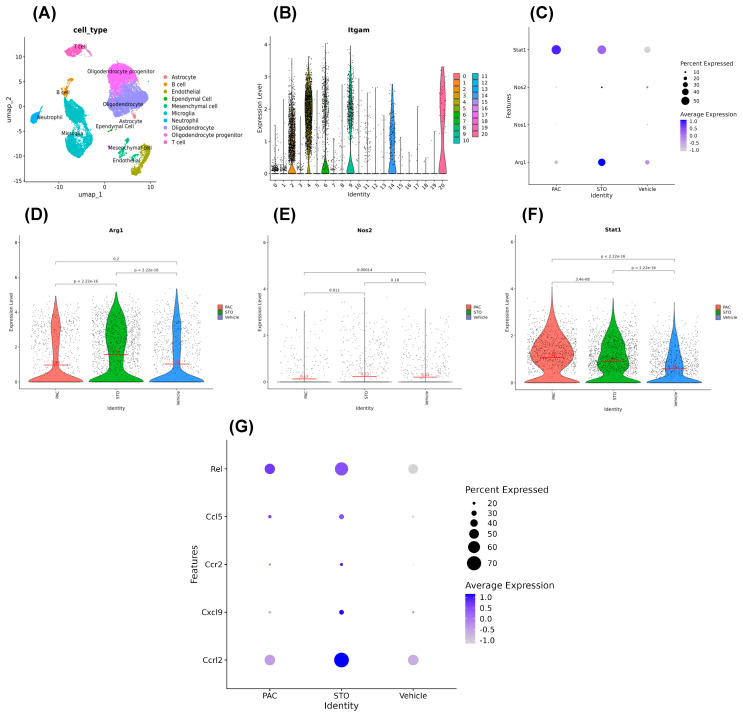
Message levels obtained from scRNA-seq do not change parallel to the protein levels for Arg1, iNOS, and STAT1. (**A**) UMAP clusters from scRNA-seq. (**B**) ITGAM distribution in clusters in violin plot. (**C**) Dot-plot for **stat1, nos1, nos2,** and **arg1** expression levels in different treatments in the ITGAM+ cells. (**D**–**F**) Showing the **arg1, nos2,** and **stat1** expression level in ITGAM+ cell again in more specific violin plots. (**G**) Compared to that in the vehicle-treated mice, a number of genes linked to the anti-tumor M1 state were induced in the TAMs of mice treated with STO-1, such as **ccrl2**, **cxcl9**, **ccr2**, **ccl5**, and **rel** in the ITGAM+ cells. Additionally, TAMs in the STO-1-treated mice showed a higher induction level of the **ccrl2** gene compared to that in the Pac-treated mice.

**Figure 8 cells-14-01703-f008:**
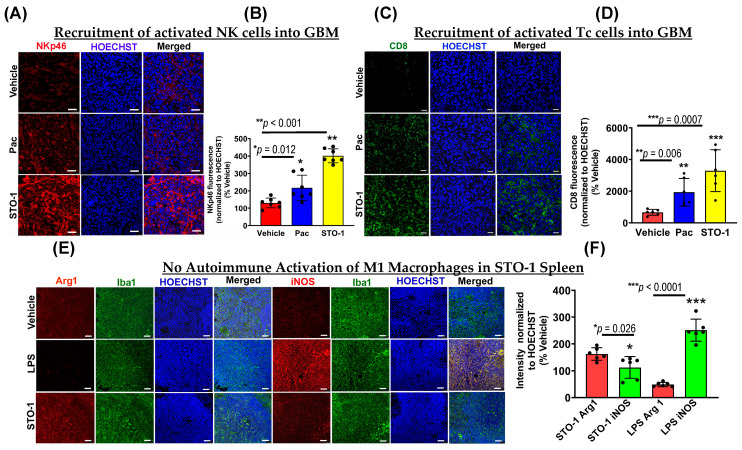
STO-1 induces recruitment of NKp46+ natural killer (NK) cells and CD8+ cytotoxic T (Tc) cells into GBM tumors, without eliciting autoimmune effects in the peripheral immune system. (**A**) and (**B**) tumors from the STO-1-treated mice as shown in Figure 3 show a large influx of NKp46+ NK cells as compared to the Vehicle-treated mice (*N* = 3 mice per treatment). (**C**,**D**) STO-1 treatment causes a dramatic influx of CD8+ Tc cells into the tumor. (*N* = 3 mice per treatment). (**E**,**F**) Cancer-free (healthy) mice were treated once with STO-1 (0.6 mg) (i.p.) or LPS (1.6 mg/Kg) (i.p.), and 24 h later, the mice were perfused under deep anesthesia with PBS and then 4% paraformaldehyde. Cryosections of the spleens were subjected to IHC to probe for Arg1 and iNOS in Iba1+ macrophages. (*N* = 3 mice per treatment). Scale bar: 25 μm.

**Figure 9 cells-14-01703-f009:**
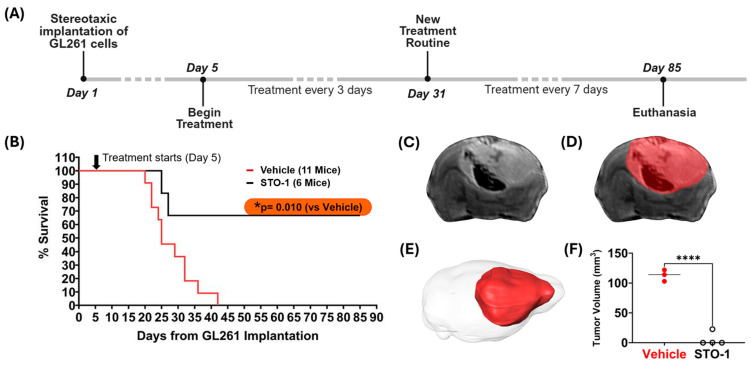
STO-1 treatment induces long-term remission of GBM in mice orthotopically implanted with murine GBM cells. (**A**) Schematic representation of the treatment timeline. Twenty-three adult C57BL/6 male mice (3–5 months old) were used. Each mouse received orthotopic implantation of 2 × 10^4^ GL261 murine GBM cells on day 1. Starting on **day 5**, mice were randomly divided into three groups and administered intravenously (i.v.) 200 µL of either Vehicle (lipid vesicles-only, V, N = 11) or STO-1-filled lipid vesicles (STO-1, 0.46 μmol, 0.6 mg/per mouse, N = 6) or liposomal Pac (0.46 μmol, 0.4 mg/per mouse, N = 6) in phosphate-buffered saline (PBS) every 72 h. For surviving mice, the treatment was spaced to 7-day intervals starting on day 31 post-implantation and continued until terminal examination following euthanasia. (**B**) Two of the six STO-1-treated mice reached the humane endpoint at days 25 and 27, respectively. However, four of the STO-1-treated mice achieved long-term survival from GBM, whereas all eleven Vehicle-treated mice reached humane endpoints within 42 days (*** *p* = 0.010**, two-tailed *t*-test with unequal variance). All six Pac-treated mice reached humane end points between days 18 and 24 (Appendix A). A subset of the moribund, Vehicle-treated mice (N = 3, days 22 to 42) and the surviving STO-1-treated mice (N = 4, day 85) were euthanized and perfusion-fixed. Their heads were decapitated to preserve tumor integrity within the skull, maintaining anatomical context for precise tumor volume assessment where applicable. (**C**) An example of a typical T2-weighted (T2-w) Fast-Spin Echo (FSE) slice image from the 3D dataset is shown, enabling full brain coverage. This example, taken from a moribund V-treated mouse euthanized at day 27, displays tumor-induced edema, evident as enhanced signal in the FSE image, while hemorrhaging appears as a dark signal. (**D**) The same slice is shown with an overlaid mask delineating the brain tumor surface and boundaries, a process performed serially across individual slices. (**E**) Full brain coverage reveals the extent of the tumor within the intracranial space as a 3D rendering. (**F**) Tumor volumes, when present, are plotted for both groups (vehicle: N = 3; STO-1: N = 4) with statistical analysis (Two-tailed *t*-test with unequal variance; ****** *p* <0.0001**).

**Figure 10 cells-14-01703-f010:**
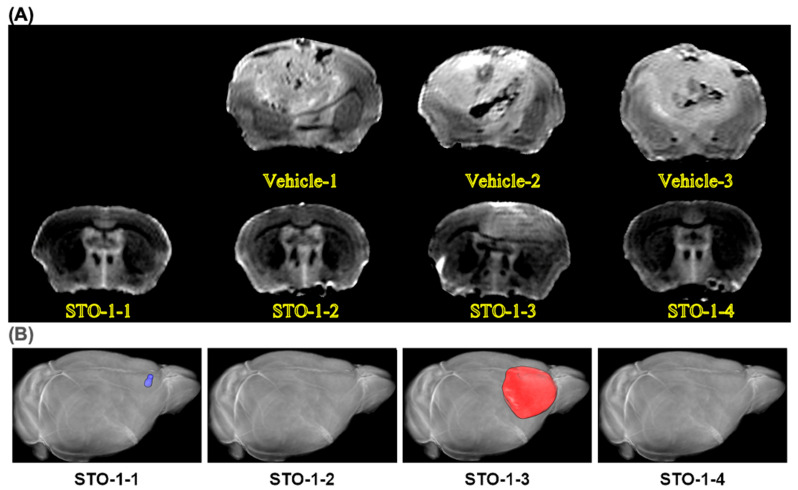
MRI images: a sampling of three Vehicle-treated mouse brains with large tumors and the four brains from the four STO-1-treated and rescued mice. (**A**) All Vehicle-treated mice developed large tumors upon reaching a moribund state, with post-implantation intervals ranging from 22 to 42 days, as depicted in the subset of brains examined via ex vivo MRI (upper row, three brains). In contrast, all four STO-1-treated mice, shown in the lower row, remained asymptomatic until the experimental endpoint at day 85 post-implantation. Notably, only one STO-1 treated mice (STO-1-3), exhibited a residual tumor that was significantly smaller than those in Vehicle-treated mice (Figure 9F). This residual tumor suggests that prolonged STO-1 treatment may be required for more effective tumor suppression. (**B**) In 3D reconstruction of the MRI scans, one of the surviving mice (STO-1-1) showed a lump of dead cells (scar tissue) that had not cleared yet (pseudo-colored blue), whereas the scar tissue was not detectable in STO-1-2 and STO-1-4 at the MRI resolution of 150 μm.

## Data Availability

We have not posted our findings in any data base, but all authors agree to freely share all data published in this article upon request.

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
