# Peer review of "Novel Taxol-Derivative, STO-1, Induces Selective Anti-Tumor Immunity and Sustained Remission of Glioblastoma Without Triggering Autoimmune Reactions"

_cells, 2025, doi:10.3390/cells14211703_

Round 1
Reviewer 1 Report
Comments and Suggestions for Authors
This is a noteworthy paper on a potent therapy using STO-1 for GBM. Hoping that this strategy can lead to a better outcome of GBM patients, let me raise some comments.
#Graphic Abstract: It is true that M1-type is ‘tumoricidal’, and M2, ‘tumor-promoting’. But as the Introduction says, M1 is ‘inflammatory’ and M2 is ‘immunosuppressive’. Thus, in addition to (or in place of) ‘tumoricidal’ and ‘tumor-promoting’, it may be better to insert such words as ‘proinflammatory’ and ‘immunosuppressive/anti-inflammatory’, respectively, for better understanding. I guess this would make this graphic abstract much more easily understandable. Not all readers are well versed in the theme (M1/M2) of this paper.
# The 4th paragraph of Introduction: STO-1 is introduced in this paragraph for the first time in this paper (the main text). In here, what it says about STO-1 is that it is ‘a novel GBM therapeutic’, which seems to have been derived from ‘a calculated choice of biocompatible molecules and strategic synthetic schemes.’ But after all, why was it STO-1 that occurred to the authors? How come STO-1? Is it a suggestion inspired from Taxol? The formal name of it is ‘Stomalignum-1’ (as in Figure 1)? The appearance of STO-1 alone here with so little of its associated information is rather abrupt, and it even says that the technology has been licensed to a commercial domain and the clinical trial using it will soon begin. If this is true, then, does it go too far to say that a paper like this is kind of redundant, just like another award for the authors with the FDA approval and clinical trial being very close at hand? I want to know the real background why the authors hit it on STO-1 for GBM, and why they had to compile this paper, which is about STO-1 for GBM in terms of its relation to intra-GBM and systemic immunological environment. One may be able to recognize these points if he/she keeps reading the paper, but it would be necessary for him/her to recognize them at the very point of this Introduction, not later in the paper.
#Treatment of Animals (page 5): It says a total of 17 mice were the subject; 6 mice were for STO-1 treatment and 11 mice were for vehicle treatment. Then where in the world were the mice treated with Pac? In line 185-186, it says intravenous treatment was done with vehicle or STO-1 or Pac. But where is Pac?
#Figure 3D
This photo is impressive. Please indicate from which case this photo was taken. Is it from a mouse treated with vehicle?
#Figure 8
I admit that the spleen is one of the centers of immune system. But I just wonder whether the spleen can really be representative of peripheral immune system or not. Is the spleen a faithful reflection of the peripheral immune system? Why the spleen of all immune systems, with, for example, the bone marrow or the thymus being at hand? Besides, what is the physiological/normal composition of M1 and M2 in the spleen? I want to know just a bit of more rationale for using the spleen for this analysis.
#3.10. Prolonged Pac Treatment Does Not Rescue Mice from GBM or Discussion
Although STO-1 is a derivative from Pac, the former, but not the latter, seems promising for GBM. I just wonder why this is so. How come a derivative is effective, but not its original? To know the possible mechanism for this kind of discrepancy between a drug and its derivative may be important when we think about the new strategy for production of new drugs. I want to know this point if it is possible.
# I have noticed some minor wording/grammatical errors that are possible. Please check the MS again.
-line 117: significant increase survival > significantly increased survival (?)
-etc.
Author Response
This is a noteworthy paper on a potent therapy using STO-1 for GBM. Hoping that this strategy can lead to a better outcome of GBM patients, let me raise some comments.
#Graphic Abstract: It is true that M1-type is ‘tumoricidal’, and M2, ‘tumor-promoting’. But as the Introduction says, M1 is ‘inflammatory’ and M2 is ‘immunosuppressive’. Thus, in addition to (or in place of) ‘tumoricidal’ and ‘tumor-promoting’, it may be better to insert such words as ‘proinflammatory’ and ‘immunosuppressive/anti-inflammatory’, respectively, for better understanding. I guess this would make this graphic abstract much more easily understandable. Not all readers are well versed in the theme (M1/M2) of this paper. Response: We have updated the Graphic Abstract accordingly.
# The 4th paragraph of Introduction: STO-1 is introduced in this paragraph for the first time in this paper (the main text). In here, what it says about STO-1 is that it is ‘a novel GBM therapeutic’, which seems to have been derived from ‘a calculated choice of biocompatible molecules and strategic synthetic schemes.’ But after all, why was it STO-1 that occurred to the authors? How come STO-1? Is it a suggestion inspired from Taxol? The formal name of it is ‘Stomalignum-1’ (as in Figure 1)? The appearance of STO-1 alone here with so little of its associated information is rather abrupt, and it even says that the technology has been licensed to a commercial domain and the clinical trial using it will soon begin. If this is true, then, does it go too far to say that a paper like this is kind of redundant, just like another award for the authors with the FDA approval and clinical trial being very close at hand? I want to know the real background why the authors hit it on STO-1 for GBM, and why they had to compile this paper, which is about STO-1 for GBM in terms of its relation to intra-GBM and systemic immunological environment. One may be able to recognize these points if he/she keeps reading the paper, but it would be necessary for him/her to recognize them at the very point of this Introduction, not later in the paper. Response: We have now addressed this important issue in the introduction in a section that is highlighted in yellow.
#Treatment of Animals (page 5): It says a total of 17 mice were the subject; 6 mice were for STO-1 treatment and 11 mice were for vehicle treatment. Then where in the world were the mice treated with Pac? In line 185-186, it says intravenous treatment was done with vehicle or STO-1 or Pac. But where is Pac? Response: We thank the reviewer for pointing out this slip. We missed the six Pac-treated mice. This error has been corrected in the revised manuscript.
#Figure 3D
This photo is impressive. Please indicate from which case this photo was taken. Is it from a mouse treated with vehicle? I was indeed from a vehicle-treated mouse. Response: Because of the use of larger number of implanted GL261 cells, coupled with a later day (day 12) of the start of STO-1 treatment and also the short (five-day) period of STO-1 treatment, the STO-1-treated mice also harbored similar-looking GBM tumors, although they were smaller than in the vehicle-treated mice.
#Figure 8
I admit that the spleen is one of the centers of immune system. But I just wonder whether the spleen can really be representative of peripheral immune system or not. Is the spleen a faithful reflection of the peripheral immune system? Why the spleen of all immune systems, with, for example, the bone marrow or the thymus being at hand? Besides, what is the physiological/normal composition of M1 and M2 in the spleen? I want to know just a bit of more rationale for using the spleen for this analysis. Response: This issue has been covered in the discussion section of the revised manuscript. The section is highlighted in yellow.
#3.10. Prolonged Pac Treatment Does Not Rescue Mice from GBM or Discussion
Although STO-1 is a derivative from Pac, the former, but not the latter, seems promising for GBM. I just wonder why this is so. How come a derivative is effective, but not its original? To know the possible mechanism for this kind of discrepancy between a drug and its derivative may be important when we think about the new strategy for production of new drugs. I want to know this point if it is possible. Response: We have now included some of the negative aspects of Paclitaxel in the Introduction. We have also included further discussion in the Discussion section.
# I have noticed some minor wording/grammatical errors that are possible. Please check the MS again.
-line 117: significant increase survival > significantly increased survival (?)
Response: Was missing the word “in” between “increase” and “survival”.

Reviewer 2 Report
Comments and Suggestions for Authors
Well conceived, executed and presented work. It explores a rather clinically relevant target in neuro-oncology, that is glioblastoma (GBM) and its suppression via a novel drug-mediated immune system activation. This approach has been successfully applied in several other types of solid malignancies and thus it is conceivable to test it in GBM.
No major issues were found in the manuscript, it is well organized, logical and content rich.
The only points where the present work could be improved are
- GBM is a cold tumor where the immune system is involved to a rather limited level and thus the role of reactivation of immune response might have in real clinical conditions not an expected impact despite of persuasive results presented in here. Authors might include in discussion this aspect and limitation. Further, GBM is a tumor with an extraordinary heterogeneity existing at all levels which explains why all current therapeutical approaches have shown only partial and time-limited success. Authors present their findings with just one cell line and a corresponding tumor which certainly do not reflect an enormous variability of the tumor in question. It should be mentioned and discussed in the text.
- All microphotographs are to have an internal scale
Author Response
Well conceived, executed and presented work. It explores a rather clinically relevant target in neuro-oncology, that is glioblastoma (GBM) and its suppression via a novel drug-mediated immune system activation. This approach has been successfully applied in several other types of solid malignancies and thus it is conceivable to test it in GBM.
No major issues were found in the manuscript, it is well organized, logical and content rich.
The only points where the present work could be improved are
- GBM is a cold tumor where the immune system is involved to a rather limited level and thus the role of reactivation of immune response might have in real clinical conditions not an expected impact despite of persuasive results presented in here. Authors might include in discussion this aspect and limitation. Further, GBM is a tumor with an extraordinary heterogeneity existing at all levels which explains why all current therapeutical approaches have shown only partial and time-limited success. Authors present their findings with just one cell line and a corresponding tumor which certainly do not reflect an enormous variability of the tumor in question. It should be mentioned and discussed in the text.
- Response: This important point has been addressed in detail in the revised discussion.
- All microphotographs are to have an internal scale
- Response: Now all the panels have scale bars in the revised manuscript.

Reviewer 3 Report
Comments and Suggestions for Authors
The current manuscript describes the synthesis and preclinical evaluation of a novel Taxol-curcumin compound, STO-1. The authors claim that STO-1 selectively reprograms tumor-associated macrophages from an M2 to an M1 phenotype in glioblastoma, thereby eliciting selective anti-tumor immunity within the glioblastoma without inducing systemic autoimmune effects. To support this, the authors employ a syngeneic GL261 mouse model of glioblastoma to test the in vivo efficacy of STO-1 delivered in a liposomal formulation. Mechanistic studies include single-cell RNA sequencing, flow cytometry, and immunohistochemistry to evaluate macrophage polarization, immune cell recruitment, and STAT1/STAT3 signaling. Here they found that STO-1 reaches the brain post-IV administration and inhibits GL261 glioblastoma cells in vitro comparably to paclitaxel. In a syngeneic GBM mouse model, STO-1 treatment significantly reprogrammed TAMs, increased STAT1 activation, and suppressed STAT3 signaling. Further administration of STO-1 induced tumor infiltration by NK and CD8+ T cells.
This manuscript presents a novel, rationally designed immunotherapeutic agent targeting glioblastoma through tumor-specific macrophage modulation. The current work is appropriate in addressing a major limitation of GBM therapies, which either involve immune evasion or systemic toxicity. However, despite the promising results, the manuscript has several major limitations related to mechanistic depth, reproducibility, and translational clarity that warrant substantive revisions.
- The concept of selective TAM reprogramming to avoid systemic inflammation is compelling and experimentally validated, and the formulation as a liposomal IV agent and discussion of IND preparation is a valuable translational component.
- The selectivity mechanism to address why M1 is suppressed in the spleen but induced in the tumor remains speculative and not explained.
- The transcriptomic data presented contradict protein-level findings (e.g., no significant change in Arg1 or Nos2 mRNA).
- The efficacy of STO-1 is compared primarily to paclitaxel. Additional controls (e.g., curcumin alone, combination vs. conjugate) would strengthen the conclusions.
- Lack of comparison to either chemo or ICB limits its translational positioning.
- While peripheral autoimmunity (spleen) is assessed, broader organ toxicity and clinical parameters (e.g., weight loss, liver/kidney function) are not reported.
- Method details are replicable; formulation and dosage calculation are well explained. It lacks pharmacokinetic data beyond brain concentration at 15–60 min and has no justification for the selection of dose frequency (every 72 hours).
- Results sections are generally clear with logical progression, but quantitative data often lack error bars, n values, or indication of replicates.
- Even though the study focuses heavily on TAMs, NK, and CD8⁺ T cells, it would be better to provide a complete immunological profile including other immune populations (e.g., dendritic cells, Tregs, B cells).
- While the authors speculate that STO-1 acts via STAT1 induction independently of STAT3 inhibition, this remains correlative. Functional experiments (e.g., using STAT1/STAT3 inhibitors or knockouts) are not included.
- Figure 4 flow data needs to be reanalyze with flow jo or related software to present it in a well format.
- In Figure 7, UMAP plots lack labeled clusters or annotations for cell types.
- Even though the discussion provides thoughtful speculation on mechanisms and clinical translation, it does not adequately address the scRNA-seq discrepancies. Further validation or more information on safety and regulatory concerns in the discussion would be beneficial.
- Line 45: "Minimal residual disease" — please specify how this was defined/measured in MRI.
- Line 81–82: Consider simplifying: "Expression of iNOS and IL-12 are key M1 markers."
- Line 103–106: The sentence is complex; consider splitting or rephrasing for clarity.
- Figure 4C: Include error bars and exact p-values on bar graphs.
- Figure 7: UMAP lacks cluster labels; axes are not annotated; dot plots should indicate % of expressing cells.
Author Response
The current manuscript describes the synthesis and preclinical evaluation of a novel Taxol-curcumin compound, STO-1. The authors claim that STO-1 selectively reprograms tumor-associated macrophages from an M2 to an M1 phenotype in glioblastoma, thereby eliciting selective anti-tumor immunity within the glioblastoma without inducing systemic autoimmune effects. To support this, the authors employ a syngeneic GL261 mouse model of glioblastoma to test the in vivo efficacy of STO-1 delivered in a liposomal formulation. Mechanistic studies include single-cell RNA sequencing, flow cytometry, and immunohistochemistry to evaluate macrophage polarization, immune cell recruitment, and STAT1/STAT3 signaling. Here they found that STO-1 reaches the brain post-IV administration and inhibits GL261 glioblastoma cells in vitro comparably to paclitaxel. In a syngeneic GBM mouse model, STO-1 treatment significantly reprogrammed TAMs, increased STAT1 activation, and suppressed STAT3 signaling. Further administration of STO-1 induced tumor infiltration by NK and CD8+ T cells.
This manuscript presents a novel, rationally designed immunotherapeutic agent targeting glioblastoma through tumor-specific macrophage modulation. The current work is appropriate in addressing a major limitation of GBM therapies, which either involve immune evasion or systemic toxicity. However, despite the promising results, the manuscript has several major limitations related to mechanistic depth, reproducibility, and translational clarity that warrant substantive revisions.
- The concept of selective TAM reprogramming to avoid systemic inflammation is compelling and experimentally validated, and the formulation as a liposomal IV agent and discussion of IND preparation is a valuable translational component.
- The selectivity mechanism to address why M1 is suppressed in the spleen but induced in the tumor remains speculative and not explained. Response: Our earlier studies with curcumin (CC) being the pioneering explorations in using an anti-inflammatory substance (curcumin) in inducing the M2®M1 transformation in the TAMs, we still cannot answer this question. We suspect that the GBM-released cytokines create this dichotomy. We have mentioned some GBM-macrophage coculture studies that we will conduct in future to answer this crucial question. This has been mentioned in the discussion.
- The transcriptomic data presented contradict protein-level findings (e.g., no significant change in Arg1 or Nos2 mRNA). Response: More explanation has been provided in the Discussion section of the revised manuscript.
- The efficacy of STO-1 is compared primarily to paclitaxel. Additional controls (e.g., curcumin alone, combination vs. conjugate) would strengthen the conclusions. Response: With reference our prior publications using curcumin, more explanation has been provided in the revised manuscript.
- Lack of comparison to either chemo or ICB limits its translational positioning. Response: We could not find any studies involving immune checkpoint blockers using the GL261-implanted syngeneic mouse model. However, other thorough studies using both protein analysis as well as scRNA-seq have been conducted to characterize this model. This important issue has been addressed in the Discussion section of the revised manuscript.
- While peripheral autoimmunity (spleen) is assessed, broader organ toxicity and clinical parameters (e.g., weight loss, liver/kidney function) are not reported. Response: Since the spleen is a major reservoir of macrophages, we used the spleen. More explanation has been provided in the revised manuscript in Discussion (highlighted in Yellow). Our humane end point for euthanasia has been described in manuscript in the methods section and legends. In the Discussion, we have mentioned about broader organ analyses that will be conducted in future.
- Method details are replicable; formulation and dosage calculation are well explained. It lacks pharmacokinetic data beyond brain concentration at 15–60 min and has no justification for the selection of dose frequency (every 72 hours). Response: Explanation has been provided in the revised manuscript in a highlighted section within Methods.
- Results sections are generally clear with logical progression, but quantitative data often lack error bars, n values, or indication of replicates. Response: Such details were included in the Figure legend, but now we have included the p values in the graphs but left the information “N = 3” in the legend in order avoid crowding.
- Even though the study focuses heavily on TAMs, NK, and CD8⁺ T cells, it would be better to provide a complete immunological profile including other immune populations (e.g., dendritic cells, Tregs, B cells). Response: None of the many publications on GL261 tumors provide information about Tregs, B cells, and dendritic cells. However, MHC I is expressed, but not MHC II and also factors that help in Tc recognition. We have added such information in a highlighted section in Discussion.
- While the authors speculate that STO-1 acts via STAT1 induction independently of STAT3 inhibition, this remains correlative. Functional experiments (e.g., using STAT1/STAT3 inhibitors or knockouts) are not included. Response: No microglia/macrophage-selective knockout of STAT3 or STAT1 has been reported. Whole animal knockout will not help in our studies. We have discussed using GBM-macrophage coculture studies while using selective blockers of STAT3 and STAT1.
- Figure 4 flow data needs to be reanalyze with flow jo or related software to present it in a well format. Response: An updated figure has been included now.
- In Figure 7, UMAP plots lack labeled clusters or annotations for cell types. Response: An updated figure has been included with cluster labels.
- Even though the discussion provides thoughtful speculation on mechanisms and clinical translation, it does not adequately address the scRNA-seq discrepancies. Further validation or more information on safety and regulatory concerns in the discussion would be beneficial. Response: We have now replaced our earlier explanation with a new paragraph that has been highlighted in yellow in Discussion.
- Line 45: "Minimal residual disease" — please specify how this was defined/measured in MRI. Response: Statistical analysis demonstrated a highly significant p value between the vehicle tumor sizes as determined by rigorous analysis of 3D MRI images and calculated tumor sizes varied from zero in three brains and a small tumor volume in one asymptomatic mouse. Furthermore, in this study, we euthanized the surviving, asymptomatic mice at day 85. At this point we observed a thin layer of GBM cells still existing in one mouse, strongly indicating that further prolonging our treatment would have eliminated these residual GBM cells.
- Line 81–82: Consider simplifying: "Expression of iNOS and IL-12 are key M1 markers." Response: Has been modified.
- Line 103–106: The sentence is complex; consider splitting or rephrasing for clarity. Response: This part has been re-written.
- Figure 4C: Include error bars and exact p-values on bar graphs. Response: We have updated the figures
Figure 7: UMAP lacks cluster labels; axes are not annotated; dot plots should indicate % of expressing cells. Response: The cluster labels have been updated. The dot plots already had “% expressing cells”.

Round 2
Reviewer 3 Report
Comments and Suggestions for Authors
Authors have answered all my questions.